# Taekwondo Enhances Cognitive Function as a Result of Increased Neurotrophic Growth Factors in Elderly Women

**DOI:** 10.3390/ijerph16060962

**Published:** 2019-03-18

**Authors:** Su-Youn Cho, Hee-Tae Roh

**Affiliations:** 1Department of Taekwondo, Youngsan University, Yangsan-si 50510, Korea; csy@swu.ac.kr; 2Department of Physical Education, College of Arts and Physical Education, Dong-A University, Busan 49315, Korea

**Keywords:** taekwondo, physical activity, neurotrophic factors, cerebral blood flow, cognition, elderly

## Abstract

The purpose of this study was to investigate the effects of regular taekwondo (TKD) training on physical fitness, neurotrophic growth factors, cerebral blood flow (CBF) velocity, and cognitive function in elderly women. Thirty-seven women aged 65 or older were randomly assigned to either TKD (*n* = 19) or control (*n* = 18) group. TKD training was performed at 50–80% maximum heart rate (HRmax) for 60 min, five times per week for 16 weeks. All participants underwent the following examinations before and after the intervention: Senior Fitness Test; serum levels of neurotrophic growth factors, including brain-derived neurotrophic factor (BDNF), vascular endothelial growth factor (VEGF), and insulin-like growth factor-1 (IGF-1); systolic, diastolic, and mean blood flow velocity and pulsatility index of the middle cerebral artery using Doppler ultrasonography; Mini-Mental State Examination for dementia screening (MMSE-DS); and Stroop Color and Word Test (word, color, and color-word). In the TKD group, lower body strength and flexibility, aerobic endurance levels, BDNF, VEGF, and IGF-1 serum levels as well as the color-word test scores were significantly increased after as compared to before the intervention (*p* < 0.05). No statistically significant differences were found in cerebral blood flow velocities and the MMSE-DS score (*p* > 0.05). These findings suggest that regular TKD training may be effective in improving not only fitness but also cognitive function in elderly women. The latter effect may be due to increased neurotrophic growth factor levels.

## 1. Introduction

Neural plasticity refers to the brain’s capacity for life-long structural and functional changes in reaction to the environment and an individual’s experiences. However, the ability of the brain to adapt to the environment gradually decreases with age, eventually causing a decline in brain function. Elderly people tend to show lower reaction speed and accuracy than younger adults in tests that measure perceptual speed, memory, and decision-making and multitasking ability, while simultaneously experiencing a cognitive decline [1,2]. With rapidly growing elderly populations worldwide over recent years, various studies have been conducted with the aim to delay this age-related cognitive decline. Attention has been particularly focused on studies examining the correlation between cognitive function and exercise, including regular physical activity, in elderly people [3,4,5,6]. Yaffe et al. [5] reported that elderly people who do not regularly participate in physical activity (walking in their study) are more likely to experience cognitive decline than elderly people who are physically active, while Laurin et al. [6] suggested that elderly people who exercise more have an appreciably lower risk of brain disease associated with cognitive dysfunction, such as dementia. According to a meta-analysis of the relationship between physical fitness and cognitive function, aerobic training has a positive effect on maintaining cognitive function in elderly people, and aerobic exercise combined with muscle strengthening or flexibility training can be even more effective in delaying cognitive decline [7,8].

The mechanism by which exercise improves brain function, including cognitive function, is not clearly understood at this time. It has been suggested that neurotrophic and angiogenic factors such as brain-derived neurotrophic factor (BDNF), vascular endothelial growth factor (VEGF), and insulin-like growth factor-1 (IGF-1), play an important role [9,10]. The expression of BDNF, which facilitates neurogenesis, is increased by exercise [10], and BDNF is known to be involved in enhancing the survival of progenitor cells that have the potential to differentiate into either nerve or glial cells and to directly promote the differentiation of progenitor cells into nerve cells [11,12]. Various studies also found that the expression of VEGF and IGF-1 is increased by regular exercise training and acute exercise, suggesting that they are implicated in maintaining and enhancing brain function [13,14,15,16]. VEGF is involved in the division and survival of endothelial cells, regulates angiogenesis, and stimulates neurogenesis [15]. Similarly, IGF-1 is also involved in the growth and differentiation of neurons, as well as in neuroprotection, regeneration, and functional plasticity of the brain [16,17].

Some studies reported that decreased cerebral blood flow (CBF) is associated with cognitive decline [18,19], and resting CBF has been found to decrease with aging [20]. A decrease in CBF velocity was identified in patients with Alzheimer’s disease [19,21] and reported to be associated with cognitive decline in non-demented subjects [18,22]. Accordingly, CBF velocity has been proposed as a potential biomarker for diagnosing Alzheimer’s [19]. At the same time, it has been suggested that beneficial effects on cognitive function may be expected from exercise training since it can increase CBF velocity [23]. Ainslie et al. [24] used transcranial Doppler (TCD) ultrasonography in 307 healthy males aged 18–79 years to measure blood flow velocity in their middle cerebral artery (MCAv). They showed that endurance-trained males (*n* = 154) had significantly higher MCAv (approximately 17%) than healthy sedentary males (*n* = 153) and concluded that regular aerobic endurance training is associated with higher MCAv. In addition, Anazodo et al. [25] tested the effects of a six-month aerobic exercise-based cardiac rehabilitation program on resting CBF in coronary artery disease patients and reported that increased CBF in the bilateral anterior cingulate region is responsible for cognitive processing. Chapman et al. [26] reported that three months of supervised aerobic exercise training resulted in increased CBF in the anterior cingulate region and improved memory performance in cognitively healthy sedentary adults.

Taekwondo (TKD), a form of martial arts that originated in Korea, is a popular sport with currently about 80 million participants in 180 countries throughout the world, a number that continues to grow [27,28]. Studies have reported that regular TKD training is effective in enhancing physical fitness, such as aerobic capacity and flexibility, as well as in improving body composition [27]. In addition, it has been suggested that a positive effect on brain function may be expected from TKD training. In a recent study by Roh et al. [28], 16-week TKD training was found to be effective in improving the cognitive function of children in the growing stage. Kim et al. [29] performed functional magnetic resonance imaging of the brain in children and found that regular TKD training significantly improved brain connectivity. Oztasyonar [30] found that athletes generally show significantly higher basal BDNF levels than sedentary subjects but that TKD training induces a greater increase in pre- and post-training BDNF levels than boxing and middle- and long-distance running.

However, it is still unclear how exactly TKD training induces structural and functional changes in the brain. Most of the studies that investigated changes in brain function associated with TKD training [28,29,30] have the limitation that their study populations consisted of children or healthy young adults. There is only very limited evidence on the effect of TKD in elderly people who experience a cognitive decline with aging. Accordingly, the objective of the present study was to investigate the effects of TKD training on the physical fitness of cognitively healthy, sedentary elderly women and to identify changes in their cognitive function by performing established cognitive function tests and by measuring CBF velocity in the middle cerebral artery (MCA) and peripheral neurotrophic growth factor levels.

## 2. Methods

### 2.1. Participants

The participants in the present study consisted of 40 healthy women aged 65 years or older who understood the significance of the experiment and volunteered to participate. Twenty women each were randomly allocated to either the taekwondo (TKD) or the control (CON) group. The selection criteria were as follows: (1) not participating in any regular exercise program; (2) completion of at least nine years of school (middle school graduate or above); (3) normal cognitive status (Mini-Mental State Examination for dementia screening [MMSE-DS] test score of ≥25 points); (4) no history of musculoskeletal, neurological, and/or psychiatric disorders; and (5) not taking drugs such as antihypertensives and statins. Among them, three participants (one in the TKD group and two in the CON group) dropped out for personal reasons. Consequently, 19 women in the TKD group and 18 women in the CON group were selected as the final participants for the study. The two groups did not significantly differ in any of the other characteristics that were recorded (Table 1).

The study purpose, method, and procedures were explained to all participants, and they signed a consent form that included information such as the option of voluntary withdrawal from participation. The study protocol was approved by the Institutional Review Board of Youngsan University (IRB No: YSUIRB-201709-BR-009-02).

### 2.2. Procedure

The study was conducted in the following order: participant selection, basic tests, pre-intervention tests, and post-intervention (16-week) tests. As part of the basic tests performed, inquiries were made to investigate all factors that may affect the variables in the study, including musculoskeletal, neurological, and/or psychiatric disorder status; participation in regular exercise programs; education level; and cognitive status. In addition, height and weight were measured and the body mass index (BMI) was calculated.

Subsequently, the TKD group members participated in a 16-week TKD training program as described below, whereas the CON group members maintained their activities of daily living without any intervention.

The pre- and post-intervention tests consisted of the Senior Fitness Test (SFT), CBF velocity measurement, cognitive function assessment, and blood collection for measurement of neurotrophic growth factor serum levels.

### 2.3. Senior Fitness Test

The SFT, developed by Rikli and Jones [31], was used to measure the physical fitness of the participating women. The SFT manual [32] was referenced for the specific measurement methods used. The SFT items selected for this study were as follows: 30 s chair stand test and 30 s arm curl test to measure lower and upper extremity muscle strength; chair sit-and-reach and back scratch test to measure flexibility; 2.44 m up-and-go test to assess agility and dynamic balance; and 2 min step test to measure aerobic endurance.

### 2.4. Blood Sampling and Analyses

Using a 21-gauge needle and Becton Dickinson Vacutainer (SST) tubes, 5 mL of blood were collected from the cardinal veins in the forehand of each participant before and after the 16-week intervention. The blood samples were centrifuged at 3000 rpm for 15 min and then stored at −80 °C until they were analyzed. The analysis of the serum levels of BDNF, VEGF, and IGF-1 was performed using a sandwich enzyme-linked immunosorbent assay (ELISA). For BDNF, we used the commercially available human BDNF Kit (#DBD00; R&D Systems, Minneapolis, MN, USA, datasheet available at https://resources.rndsystems.com/pdfs/datasheets/dbd00.pdf); for VEGF, the human VEGF Kit (#DVE00; R&D Systems, Minneapolis, MN, USA, datasheet available at https://resources.rndsystems.com/pdfs/datasheets/dve00.pdf); and for IGF-1, the human IGF-1 DuoSet (#DY291; R&D Systems, Minneapolis, MN, USA, datasheet available at https://resources.rndsystems.com/pdfs/datasheets/dy291.pdf). A micro-plate reader (Emax; Molecular Devices, San Jose, CA, USA) was used to measure absorbance at 450 nm for quantification.

### 2.5. Cerebral Blood Flow Velocity Measurements

CBF velocities were measured with reference to a previous study by Aaslid et al. [33]. The systolic flow velocity (SFV), diastolic flow velocity (DFV), and mean flow velocity (MFV) of the middle cerebral arteries (MCAs) were measured through the right transtemporal window using a 2-MHz pulsed Doppler ultrasound system with the subject in the supine position after 15 min of rest. The Doppler probe was secured with a headband device to maintain an optimal insonation position and angle throughout the protocol. The pulsatility index (PI) was calculated using the following formula: PI = SFV − DFV/MFV. To increase inter-rater reliability, the measurement was performed by a radiology technician who was licensed by a government authority.

### 2.6. Cognitive Function Measurements

For cognitive function assessment, the Korean versions of the MMSE-DS [34] and the Stroop Color and Word Test [35] were used, the validity and reliability of which were verified by Kim et al. [34]. The original MMSE is a widely used tool to accurately measure and screen cognitive impairment [36]. The MMSE-DS, a standardized diagnostic tool for dementia (cognitive impairment) with a raw score ranging between 0 and 30 points, was developed to increase the accuracy and reliability of the test in a Korean population [34]. The MMSE-DS assesses the cognitive state of the respondent using 19 items, including time and place orientation, memory registration and recall, attention and concentration ability (calculation/subtraction), naming, repetition and pronunciation, praxis (the ability to synthesize and sequence motor tasks), interlocking pentagon drawing, comprehension, judgment, reason for cleaning, and interpretation of proverbs. The MMSE-DS uses different cut-off values based on gender, age, and education level of respondents. All participants in this study showed normal levels of cognitive function.

The Stroop Color and Word Test comprises word reading (Word), color reading (Color), and color-word reading (Color-Word) parts. Each part contained 100 items arranged in 5 columns and 20 rows. In the word reading part, the words for the colors “Red, Blue, Green” were written in black ink. In the color reading part, “XXXX” was written in blue, red, and green color ink, which was randomly assigned. In the color-word part, participants had to read 100 words printed in a different color than the color name. All participants were given 45 s for each part, and the correct performances were counted.

### 2.7. Taekwondo Training Intervention

TKD training consisted of a session of 60 min: a total of 10 min for the warm-up and cool-down through stretching, and 50 min for the main exercise at 50–80% maximum heart rate (HRmax). This session was performed five times per week for a total of 16 weeks. All training was conducted by a TKD expert instructor who demonstrated the exercises and coached participants. The main exercise consisted of 5 min of five basic TKD movements (stance, block, punch, strike, and thrust), 10 min of Poom-sae Taegeuk chapter 1–4, 10 min of kicking sessions with basic kicking and steps as well as mitt kicks, and 15 min of Taekwon gymnastics. The exact TKD training session schedule is shown in Table 2.

### 2.8. Statistical Analysis

Data are expressed as means and standard deviations. All analyses were performed using IBM SPSS Statistics for Windows version 24.0 (IBM Corp., Armonk, NY, USA). Two-way repeated measures analysis of variance (ANOVA) was performed to assess potential time (before and after intervention) and group (between the TKD and CON groups) differences in each variable. Both independent and dependent *t*-tests were conducted to identify statistically significant interactions. Statistical significance (*α*) was set at 0.05.

## 3. Results

### 3.1. Post-Intervention Changes in Senior Fitness Test

Changes in the SFT parameters in the TKD and CON groups before and after the 16-week intervention are shown in Table 3. Following intervention, repeated ANOVA measures demonstrated a significant difference across time by group interaction for 30 s chair stand (*F* = 40.092, *p* < 0.001), chair sit-and-reach (*F* = 9.588, *p* = 0.004), and 2 min step (*F* = 5.488, *p* = 0.025) tests. The post hoc analysis revealed significant increases in the 30 s chair stand, chair sit-and-reach, and 2 min step test scores (*p* < 0.05) in the TKD group after the intervention, but no significant differences between the 30 s arm curl, back scratch, and 2.44 m up-and-go test scores (*p* > 0.05) before and after the intervention. No significant differences between any of the SFT scores before and after the intervention were observed in the CON group (*p* > 0.05).

### 3.2. Post-Intervention Changes in Serum Neurotrophic Growth Factors

Changes in the serum levels of the neurotrophic growth factors in the TKD and CON groups before and after the 16-week intervention are shown in Table 4. Following intervention, repeated ANOVA measures demonstrated a significant difference across time by group interaction for BDNF (*F* = 5.320, *p* = 0.027), VEGF (*F* = 8.505, *p* = 0.006), and IGF-1 (*F* = 8.422, *p* = 0.006) levels. Post hoc analysis revealed significant increases in BDNF, VEGF, and IGF-1 levels (*p* < 0.05) in the TKD group after the intervention, whereas no significant differences were observed in the serum levels before and after the intervention in the CON group (*p* > 0.05).

### 3.3. Post-Intervention Changes in Cerebral Blood Flow Velocities

Changes in the CBF velocities in the TKD and CON groups before and after the 16-week intervention are shown in Table 5. Following intervention, repeated ANOVA measures demonstrated no significant difference across time by group interaction for SFV, DFV, MFV, and PI (*p* > 0.05).

### 3.4. Post-Intervention Changes in Cognitive Function

Changes in the cognitive function parameters in the TKD and CON groups before and after the 16-week intervention are shown in Table 6. Following intervention, repeated ANOVA measures demonstrated a significant difference across time by group interaction for Color-Word component (*F* = 5.764, *p* = 0.022) score. Post hoc analysis revealed a significant increase in the Color-Word component score in the TKD group (*p* < 0.05) after the intervention, but no significant differences were observed in the MMSE-DS, Word, and Color test scores (*p* > 0.05). No significant differences between any of the test results before and after the intervention were observed in the CON group (*p* > 0.05).

## 4. Discussion

Regular physical activity and exercise are considered to be among the most reliable methods for “healthy aging” [37]. It has been proven that long-term regular exercise promotes a healthy body composition, improves cardiovascular, respiratory, and musculoskeletal functions, and prevents injuries in elderly people [38,39]. TKD training has been shown to be effective in improving the fitness of healthy subjects, including their body composition, aerobic capacity, muscle strength, agility, and flexibility [27,40]. In the present study, the SFT was performed to test the effects of TKD training on muscle strength, flexibility, agility, dynamic balance, and aerobic endurance of elderly women. The results show that the 30 s chair stand and chair sit-and-reach test scores increased significantly after TKD training. This suggests that TKD training is effective in improving lower extremity muscle strength and flexibility in elderly women, probably as a result of the repeated training of TKD-specific motions, such as various stances and kicks. The muscles in the lower limbs play an important role in explosive kicking and maintaining stances, and it has been suggested that TKD training improves muscle strength through these bodyweight resistance exercises [27]. Moreover, Toskovic et al. [41] compared muscle strength between beginners and black belt practitioners of TKD and showed significantly higher lower body muscle strength in the black belt practitioners, which supports the findings of this study. TKD training has also been reported to benefit flexibility in older subjects [42,43]. Kick motions during TKD training require a high level of lower body flexibility. Brudnak et al. [42] reported that 17 weeks of TKD training improved the trunk flexibility of healthy elderly subjects (mean age, 71 years), while Cromwell et al. [43] reported that the results in the sit-and-reach test for flexibility increased significantly after TKD training. These findings reflect the potential of TKD training to slow down the biological deterioration of body flexibility.

The 2 min step test score also increased significantly in this study. This represents improved aerobic endurance capacity and indicates that the exercise frequency, intensity, and duration of the TKD training intervention were sufficient to improve cardiorespiratory fitness. The American College of Sports Medicine guidelines [44] recommend dynamic activities that recruit large muscle groups (3–5 sessions of exercise lasting 20–60 min at an intensity of 55–90% HRmax) to improve cardiopulmonary endurance. Studies have measured heart rates of 64.7–69.4% HRmax while participants were performing punches and kicks during TKD training, and of 74.7–81.4% HRmax while performing TKD basic techniques and forms [45]. Moreover, Fong and Ng [27] critically reviewed 23 papers on the effects of TKD training on physical fitness and showed that TKD training is effective in improving aerobic fitness, which supports the findings in the present study.

One of the mechanisms leading to improved brain function through regular exercise involves neurotrophic growth factors, such as BDNF, VEGF, and IGF-1, the expression of which is increased by exercise [9,10]. In particular, BDNF helps the growth and survival of various nerve cells and is a key neurotrophic factor that regulates synaptic plasticity. The expression of BDNF is increased not only by acute and chronic exercise, but also by aerobic and resistance exercise [46,47], and the degree of its expression is dependent on the intensity of the exercise [48]. The results of this study showed a significant increase in the serum levels of BDNF, VEGF, and IGF-1 after TKD training, consistent with previous findings that TKD training increased BDNF levels [30] and increased BDNF, VEGF, and IGF-1 levels after TKD training in children [49]. Some previous exercise-related studies have reported that high neurotrophic growth factor levels are closely associated with improved fitness [50,51]. The increased neurotrophic growth factor levels found in this study likely also had a major impact on increased fitness, such as muscle strength and aerobic capacity. A study by Kim et al. [50] reported that increased BDNF serum concentrations due to exercise were significantly correlated with improved cardiovascular fitness and leg strength, and resting serum IGF-1 concentrations were positively correlated with aerobic fitness levels [51]. Moreover, IGF-1 has been reported to be an upstream factor in the signaling pathway that regulates BDNF expression [52] and that it is also involved in VEGF expression [53]. Carro et al. [52] proved that the injection of IGF-1 into the peripheral vessels of Wistar rats can increase BDNF expression in the hippocampus [52], while Lopez-Lopez et al. [53] reported that systemic injection of IGF-1 effectively stimulates angiogenesis in the brain through regulation of VEGF.

Owing to advances in functional neuroimaging analysis techniques, including magnetic resonance imaging (MRI), functional MRI, positron emission tomography, and magnetoencephalography, morphometric analysis of the brain has become easier and allows the non-invasive investigation of structural and functional changes in the brain that appear during the aging process [54,55]. TCD does not provide a full view of the entire cerebral vasculature, but it offers the advantage of being able to non-invasively measure the intracranial hemodynamics in real time without using a contrast agent or radiation exposure. Accordingly, recent studies have used TCD to test the effects of exercise on changes in CBF [56,57]. Ainslie et al. [24] suggested that regular endurance training can increase the flow velocity in the MCA of healthy male subjects. The present study did not find significant differences in the SFV, DFV, MFV, and PI of the MCA after TKD training. It may be necessary for future studies to conduct tests considering the duration of exercise. In a study, regular aerobic exercise was shown to increase the flow velocity in the MCA of healthy subjects, but at least two years of regular exercise were required for this effect to occur [24]. Murrell et al. [58] measured MCAv after an aerobic exercise intervention in young (23 ± 5 years) and older adults (63 ± 5 years) and found that there was no significant difference in resting MCAv between the two groups.

The fact that participation in regular exercise contributes to maintaining and improving cognitive function, and that improved fitness plays a definitive role in delaying cognitive decline caused by aging, has been proven in many studies [3,4,5,6,7,8]. This study used the MMSE-DS and Stroop Color and Word Test to investigate the effects of TKD training on the cognitive function of elderly women. The results showed that there was no significant change in MMSE-DS scores after TKD training, but the ColorWord test as a subcategory of the Stroop Color and Word Test showed a significant increase. These findings suggest that TKD training may be effective in improving cognitive function and the improved aerobic fitness and increase in neurotrophic growth factor levels might play a crucial role in this regard. Barnes et al. [59] reported a high correlation between cardiorespiratory fitness levels and cognitive function in 349 subjects aged 55 years or older and found that peak oxygen consumption measurements were an important indicator for predicting future cognitive function outcomes. More recently, a study in adults aged 55 years or older (mean age, 61.0 ± 6.0 years, *n* = 4931) showed an association between cardiorespiratory fitness and cognitive function [60]. Specifically, when the prevalence of cognitive impairment was related to different levels of cardiorespiratory fitness, 33% of participants in the quintile with the lowest fitness level (*n* = 326) showed cognitive impairment, whereas only 19.3% of those in the quintile with the highest fitness level (*n* = 1938) did, which was a statistically significant difference. Consequently, elderly people with low cardiorespiratory fitness levels should aim to improve their fitness, while those with moderate to high levels should focus on maintaining those to protect their cognitive function.

This study has some limitations. First, as a single-center study, the study population was small and limited to elderly women only. It is necessary to conduct future studies including elderly men and bigger populations. Second, the study measured CBF velocity only in the MCA. Future studies should measure blood flow in various parts of the brain that are responsible for cognitive function, and functional neuroimaging should be used to illustrate the structural and functional changes in the brain with exercise. Third, it cannot be assumed that all changes in the variables observed in this study were caused by increased physical activity or unique effects of TKD only. Future studies must include aerobic exercise, which has been proposed to be effective in improving cognitive function, for comparative analysis.

## 5. Conclusions

In conclusion, the findings of this study suggest that participation in regular TKD training may be effective in improving the cognitive function of elderly women by increasing their fitness, such as aerobic capacity, and lower extremity muscle strength and flexibility, as well as by inducing an increase in neurotrophic growth factor levels.

## Figures and Tables

**Table 1 ijerph-16-00962-t001:** Participant characteristics.

Variables/Group	TKD (*n* = 19)	CON (*n* = 18)	*p*-Value ^b^
Age (years)	68.89 ± 4.16	69.00 ± 4.41	0.938
Height (cm)	153.29 ± 4.91	153.75 ± 4.28	0.766
Weight (kg)	56.88 ± 6.96	56.44 ± 7.11	0.848
BMI (kg/m^2^)	24.12 ± 1.73	23.84 ± 2.47	0.705
Education level (years)	11.33 ± 2.47	11.37 ± 2.41	0.965
Cognitive status ^a^ (score)	26.89 ± 1.81	26.74 ± 1.63	0.790

Data are presented as mean ± standard deviation. TKD, taekwondo; CON, control; BMI, body mass index. ^a^ Cognitive status as determined using the MMSE-DS; ^b^
*p*-value as determined using the independent *t*-test for each of the two groups at baseline.

**Table 2 ijerph-16-00962-t002:** Taekwondo (TKD) training program.

Classification		Contents	Time
Warm-up		Stretching	10 min
Main exercise	TKD basic movement	Stance, Block, Punch, Strike, Thrust	5 min
	Poom-sae	Taegeuk 1–4 chapter	10 min
	Kicking	Front kick, Side kick, Round house kick, Downward kick, Step (forward, side, backward), Practice mitt kicking	10 min
	Taekwon gymnastics	2 music-based gymnastics	15 min
Cool-down		Stretching	10 min

**Table 3 ijerph-16-00962-t003:** Senior Fitness Test (SFT) parameters in the TKD and CON groups before and after the 16-week intervention.

Variables/Group	TKD (*n* = 19)		CON (*n* = 18)		Time × Group Interaction	
	Baseline	16 Weeks	Baseline	16 Weeks	*F*	*p*
30 s arm curl (rep.)	17.94 ± 4.33	18.06 ± 4.52	17.63 ± 4.03	17.79 ± 3.88	0.014	0.906
CV	0.24	0.25	0.23	0.22		
30 s chair stand (rep.)	12.72 ± 3.34	14.89 ± 3.36 ^#^	13.05 ± 3.70	13.16 ± 3.20	40.092	<0.001 ***
CV	0.26	0.23	0.28	0.24		
Back scratch (cm)	−5.06 ± 6.43	−4.61 ± 6.14	−3.53 ± 5.35	−3.74 ± 4.98	1.701	0.201
CV	1.27	1.33	1.52	1.33		
Chair sit-and-reach (cm)	6.56 ± 5.36	7.67 ± 4.72 ^#^	7.16 ± 4.52	6.95 ± 4.10	9.588	0.004 **
CV	0.82	0.62	0.63	0.59		
2.44 m up-and-go (s)	8.25 ± 1.65	7.95 ± 1.37	7.88 ± 1.70	7.98 ± 1.61	3.917	0.056
CV	0.20	0.17	0.22	0.20		
2 min step (rep.)	84.89 ± 17.65	91.33 ± 17.49 ^#^	85.26 ± 15.60	87.16 ± 16.39	5.488	0.025 *
CV	0.21	0.19	0.18	0.19		

Data are presented as mean ± standard deviation. CV, coefficient of variation; TKD, taekwondo; CON, control. ^#^ Compared with baseline within the group (*p* < 0.05); *** *p* < 0.001; ** *p* < 0.01; * *p* < 0.05.

**Table 4 ijerph-16-00962-t004:** Serum levels of the neurotrophic growth factors in the TKD and CON groups before and after the 16-week intervention.

Variables/Group	TKD (*n* = 19)		CON (*n* = 18)		Time × Group Interaction	
	Baseline	16 Weeks	Baseline	16 Weeks	*F*	*p*
BDNF (ng/mL)	22.61 ± 7.77	25.64 ± 7.87 ^#^	21.35 ± 7.06	21.81 ± 6.98	5.320	0.027 *
CV	0.34	0.31	0.33	0.32		
VEGF (pg/mL)	139.99 ± 42.44	146.55 ± 43.11 ^#^	137.45 ± 31.96	134.73 ± 35.56	8.505	0.006 **
CV	0.30	0.29	0.23	0.26		
IGF-1 (pg/mL)	259.19 ± 70.81	278.22 ± 69.59 ^#^	251.74 ± 74.31	252.66 ± 71.03	8.422	0.006 **
CV	0.27	0.25	0.30	0.28		

Data are presented as mean ± standard deviation. CV, coefficient of variation; TKD, taekwondo; CON, control; BDNF, brain-derived neurotrophic factor; VEGF, vascular endothelial growth factor; IGF-1, insulin-like growth factor-1. ^#^ Compared with baseline within the group (*p* < 0.05); ** *p* < 0.01; * *p* < 0.05.

**Table 5 ijerph-16-00962-t005:** Cerebral blood flow (CBF) velocities in the TKD and CON groups before and after the 16-week intervention.

Variables/Group	TKD (*n* = 19)		CON (*n* = 18)		Time × Group Interaction	
	Baseline	16 Weeks	Baseline	16 Weeks	*F*	*p*
SFV (cm/s)	60.94 ± 18.08	61.33 ± 20.33	61.47 ± 15.79	61.42 ± 16.33	0.071	0.791
CV	0.30	0.33	0.26	0.27		
DFV (cm/s)	24.39 ± 10.30	25.11 ± 11.19	25.11 ± 9.61	25.79 ± 10.57	0.001	0.974
CV	0.42	0.45	0.38	0.41		
MFV (cm/s)	39.67 ± 12.12	40.28 ± 14.45	39.74 ± 12.96	39.74 ± 13.39	0.181	0.673
CV	0.31	0.36	0.33	0.34		
PI (unit)	0.94 ± 0.12	0.93 ± 0.15	0.95 ± 0.15	0.95 ± 0.20	0.013	0.910
CV	0.13	0.16	0.15	0.21		

Data are presented as mean ± standard deviation. CV, coefficient of variation; TKD, taekwondo; CON, control; SFV, systolic flow velocity; DFV, diastolic flow velocity; MFV, mean flow velocity; PI, pulsatility index.

**Table 6 ijerph-16-00962-t006:** Cognitive function parameters in the TKD and CON groups before and after the 16-week intervention.

Variables/Group	TKD (*n* = 19)		CON (*n* = 18)		Time × Group Interaction	
	Baseline	16 Weeks	Baseline	16 Weeks	*F*	*p*
MMSE-DS (score)	26.89 ± 1.81	27.56 ± 1.58	26.74 ± 1.63	27.00 ± 1.41	3.358	0.075
CV	0.07	0.06	0.06	0.05		
Word (score)	56.44 ± 21.00	57.11 ± 18.39	55,32 ± 20.97	56.37 ± 20.66	0.030	0.863
CV	0.37	0.32	0.38	0.37		
Color (score)	35.17 ± 14.16	40.11 ± 13.83	35.63 ± 13.85	36.95 ± 13.01	3.010	0.092
CV	0.40	0.34	0.39	0.35		
Color-Word (score)	23.94 ± 10.26	28.78 ± 9.47 ^#^	24.79 ± 12.24	25.42 ± 11.95	5.764	0.022 *
CV	0.43	0.33	0.49	0.47		

Data are presented as mean ± standard deviation. CV, coefficient of variation; TKD, taekwondo; CON, control; MMSE-DS; Mini-Mental State Examination for dementia screening. ^#^ Compared with baseline within the group (*p* < 0.05); * *p* < 0.05.

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
