# Peer review of "Taekwondo Enhances Cognitive Function as a Result of Increased Neurotrophic Growth Factors in Elderly Women"

_ijerph, 2019, doi:10.3390/ijerph16060962_

Round 1

Reviewer 1 Report

Lines 98-108: were the participants taking any medications/supplements that could affect your outcome variables (i.e. hypertensive medications, statins, etc.).  If so, how did you control for this?

Lines 149-154: explain how much skill (if any) is required to operate the Doppler ultrasound system.  Did the same researcher/investigator administer the test to all participants?  If not, were you concerned about inter-rater variability?

Lines 169-173: explanation of the Stroop color-word test was difficult to follow.  Consider revising for clarity.

Lines 328-336: because of the small sample size, do you feel this affected your results, since some of the tests approached significance?  Have you validated the measures that you used to assess cerebral blood flow velocity?  These should be addressed.  You did not discuss how these findings might be applied to a diseased population.  Could TKD training be used in a diseased (i.e. Alzheimer's Disease) population and elicit positive results? 

Overall, well written paper that has practical significance. 

Author Response

Manuscript ID: ijerph-458277

Type: Article

Title: Taekwondo enhances cognitive function as a result of increased neurotrophic growth factors in elderly women

REVIEWER #1

Lines 98-108: were the participants taking any medications/supplements that could affect your outcome variables (i.e. hypertensive medications, statins, etc.).  If so, how did you control for this?

Thank you very much for your helpful comments. The study participants were healthy elderly women who did not take drugs such as antihypertensives and statins. This is stated in the section “2.1. Participants.”

Lines 149-154: explain how much skill (if any) is required to operate the Doppler ultrasound system.  Did the same researcher/investigator administer the test to all participants?  If not, were you concerned about inter-rater variability?

Thank you very much for your helpful comments. To increase inter-rater reliability, CBF was measured by a radiology technician who was licensed by a government authority. This is stated in the section “2.5. Cerebral blood flow velocity measurements.”

Lines 169-173: explanation of the Stroop color-word test was difficult to follow.  Consider revising for clarity.

Thank you very much for your helpful comments. We have revised that and have highlighted it in yellow.

Lines 328-336: because of the small sample size, do you feel this affected your results, since some of the tests approached significance?  Have you validated the measures that you used to assess cerebral blood flow velocity?  These should be addressed. 

Thank you very much for your helpful comments. We believe that the sample size was sufficient for testing the study hypotheses. However, we have pointed out a limitation that the sample population had only women. Additionally, we believe that there is no errors in the measurements because CBF velocity was measured by a radiology technician who was licensed by a government authority.

You did not discuss how these findings might be applied to a diseased population.  Could TKD training be used in a diseased (i.e. Alzheimer's Disease) population and elicit positive results?

Thank you very much for your helpful comments. Because “Poom-sae” in TKD training-specific movements is stored in the memory, we believe it would be difficult to apply it to patients with severe Alzheimer’s disease. However, the training is believed to have a positive effect on the elderly experiencing normal age-related cognitive decline.

Overall, well written paper that has practical significance.

Thank you very much for considering our research paper.

Reviewer 2 Report

This study investigated the effect of Taekwondo training on the cognitive function in elderly women in Korea. This training could enhance the lower body strength and flexibility, aerobic endurance levels, color-word test score and growth factors in elderly women. They concluded that Taekwondo training could improve the cognitive function and this effect may be due to increased neurotrophic growth factor levels. 

1. For the measured BDNF, VEGF, and IGF-1, what are the detection limit? sensitivity? range? for these cytokines?

Author Response

Manuscript ID: ijerph-458277

Type: Article

Title: Taekwondo enhances cognitive function as a result of increased neurotrophic growth factors in elderly women

REVIEWER #2

This study investigated the effect of Taekwondo training on the cognitive function in elderly women in Korea. This training could enhance the lower body strength and flexibility, aerobic endurance levels, color-word test score and growth factors in elderly women. They concluded that Taekwondo training could improve the cognitive function and this effect may be due to increased neurotrophic growth factor levels.

Thank you very much for considering our research paper.

1. For the measured BDNF, VEGF, and IGF-1, what are the detection limit? sensitivity? range? for these cytokines?

Thank you very much for your helpful comments. For readers to examine the datasheets for the variables, an online link is inserted.